# Proprietary Model of Qualification for In-Hospital Rehabilitation after COVID-19

**DOI:** 10.3390/ijerph191610450

**Published:** 2022-08-22

**Authors:** Jan Szczegielniak, Anna Szczegielniak, Jacek Łuniewski, Katarzyna Bogacz

**Affiliations:** 1Physiotherapy Department, Faculty of Physical Education and Physiotherapy, Opole University of Technology, 45-758 Opole, Poland; 2Ministry of Internal Affairs and Administration’s Specialist Hospital of St. John Paul II, 48-340 Głuchołazy, Poland; 3Department of Psychoprophylaxis, Faculty of Medical Sciences in Zabrze, Medical University of Silesia, 40-055 Katowice, Poland; 4Stobrawskie Medical Center in Kup, 46-082 Kup, Poland

**Keywords:** COVID-19, rehabilitation, physical therapy

## Abstract

Background: Since the beginning of the SARS-CoV-2 epidemic in Poland, 6,128,006 people have been diagnosed, of which 116,798 died. Patients who recovered from COVID-19 and require rehabilitation due to varied impairments should be provided an opportunity to participate in an individualized, complex rehabilitation program starting from acute care and being continued in the post-acute and long-term rehabilitation phase. It is recommended to offer out-patient and in-hospital rehabilitation procedures depending on the type and persistence of symptoms and dysfunctions. The aim of this paper is to present the qualification process of post-COVID19 patients for an in-hospital complex rehabilitation program developed on the basis of pulmonary physical therapy. Methods: The presented qualification program was developed on the basis of clinical experience of over 2000 patients participating in the pilot program of in-hospital rehabilitation launched in September 2020 and based on the Regulation of the Polish Minister of Health of 13 July 2020. Results: The proposed model of patients’ qualification rests on well-known and validated tools for functional assessment: exercise tolerance assessment, dyspnea intensity assessment, functional fitness assessment, assessment of arterial blood saturation, lung ventilation function assessment, assessment of long-lasting COVID-19 symptoms, and patient’s basic mental health condition. Conclusions: The proposed qualification model for the post-COVID rehabilitation program allows us to introduce adequate qualifications followed by much needed assessment of the health effects.

## 1. Introduction

According to WHO data at the beginning of August 2022, there have been 589,680,368 confirmed cases of COVID-19, including 6,436,519 deaths [1]. The latest report of the Ministry of Health on the coronavirus in Poland of 17 August 2022, states that since 4 March 2020, when the first case of infection was diagnosed, 6,128,006 people fell ill, of which 116,798 died. From 20 March 2020 to 15 May 2022, by the ordinance of the Minister of Health, a state of epidemic emergency was announced. The introduction was associated with significant limitations of daily functioning covering, among many, transportation, operation of certain state institutions and the private sector, organization of mass events and gatherings, and access to healthcare. Wide and varied changes in working conditions were also observed. There was a legal obligation in power to use personal protective equipment in public spaces [2]. Currently, about 59.4% of the population is fully vaccinated, which is lower than the average of the global population [3]. The total cost of treating COVID-19 patients is difficult to assess due to the dynamically changing valuations of the National Health Fund. Data from 2021 indicated that hospitalization related to COVID-19 treatment is estimated by the state at 430 PLN for each day of hospital stay. However, in the case of patients requiring mechanical ventilation outside the anesthesiology and intensive care unit (ICU), 1054 PLN per day. Hospitalization at the ICU is several times more expensive. Vaccinations, free for citizens, also had variable costs throughout the epidemic [4].

Scientific research published so far indicates that COVID-19 survivors may experience persistent symptoms affecting various organ systems after the acute phase of infection involving not only the respiratory system, but also cardiovascular, musculoskeletal, gastrointestinal, immunological, endocrine, and neurological systems with varied intensity and duration. Histopathological studies have shown direct and indirect damage to many organs caused by COVID-19. A recently published paper indicates more than 50 health effects observed are fatigue, headache, attention disorder, hair loss, and dyspnea as the five most common manifestations [5]. Obtained so far and still emerging evidence points that the condition, usually referred to as long COVID or post-COVID-19, may become a significant global health burden for healthcare systems across the globe. Demands for new clinical and health policy strategies to address the allocation of resources in line with the needs of the impaired mental and physical health status have been brought up. Decreased economic productivity, lower quality of life, loss of independence, and conceivably a shortened life expectancy among survivors should be considered when planning dedicated healthcare services. An integrative approach with a wide range of individually-tailored services is being advocated as a vital addition to the actions to reduce acute COVID-19 cases [6,7]. Varied systematic reviews and metanalyses do not agree on the prevalence of persistent symptoms after COVID-19 infection ranging in some study groups between 43% to 71.9% regardless of the severity of the disease (hospitalized and non-hospitalized patients) [8,9]. Assessment of this condition seems to be challenging due to a lack of shared agreement on a clinical definition and terminology including duration, agreed list of long COVID symptoms, and tools for them to be measured. Available findings have limitations as conclusions are drawn by excluding patients who have had post-COVID-19 but were not diagnosed or didn’t receive medical attention [10,11]. This appeals, especially to patients with dysautonomic problems more common among non-hospitalized COVID-19 patients as these patients, may have fewer respiratory symptoms [12]. Persistent symptoms are highly heterogenous and non-specific, multifactorial in etiology. In addition, some of the post-COVID long-term symptoms are also observed in other viral diseases, including flu [13]. Previous coronavirus outbreaks also showed similar postdischarge symptoms [14]. Researchers point out that failure to take into account the effects of psychiatric disorders such as depression and anxiety disorders may be misleading and somatization may be accounted for additional post-COVID symptoms [15].

According to WHO rehabilitation is a set of interventions designed to optimize functioning and reduce disability in individuals with health conditions in interaction with their environment and consist of different types of services, such as physical therapy, occupational therapy, or speech therapy, and can be conducted as inpatient and outpatient treatment [16]. The very basic and vital stage of any rehabilitation program for patients who have suffered from coronavirus infection is proper qualification rehabilitation model for individually-tailored services. Most of the hospital-based rehabilitation programs developed are based on the existing recommendations of cardiopulmonary physical therapy programs with additional elements of neuromuscular activities and psychosocial support, enabling appropriate and individual dosing of physical exercise adapted to the needs and capabilities of the patient [17]. The aim of these programs is mitigation of the negative effects of the disease and patients’ support in recovery to full physical functionality. Assessment of patients’ physical efficiency, including exercise desaturation, as well as their physical and mental functions is required before planning further therapeutic actions. Moreover, maintaining a safe environment for patients and health providers is an additional challenge in terms of rehabilitation services provided for patients with active COVID-19 infection is a challenge [18].

The aim of this paper is to present the qualification process of adult post-COVID-19 patients for an in-hospital uniquely defined complex rehabilitation program developed on the basis of pulmonary physical therapy. During the pilot program, the qualification program was modified and adapted to the existing functional disorders in the ranges described.

## 2. Materials and Methods

The presented qualification program was developed on the basis of clinical experience of over 2500 patients participating in the pilot program in the field of hospital rehabilitation after COVID-19 disease, launched in September 2020. The aim of the pilot program is not only to assess the effects of the rehabilitation model used and to improve the limited exercise capacity associated with impaired ventilation and dyspnea in people after coronavirus infection, but also to identify the persistent additional symptoms and the resulting rehabilitation needs related to neurological and psychiatric disorders and symptoms from the musculoskeletal system.

The research hypothesis adopted for the rehabilitation program of adult post-COVID-19 patients assumed that the use of different rehabilitation models based on the existing pulmonary rehabilitation programs and depending on the individually obtained results in the qualification process is the right procedure for this group of patients.

The program is financed under the funds provided for in the financial plan of the National Health Fund. The cost of treatment is 200 PLN per person per day. The pilot program is aimed at COVID-19 survivors who received a referral from a health insurance physician (any health insurance physician, including a primary health care physician). The diagnosis of COVID-19 was based on a clinical picture of the infection with well-defined symptoms and positive nasopharyngeal swab polymerase chain reaction test for SARS-CoV-2, serologic anti-SARS-CoV-2 antibody test, and/or typical findings on chest computed tomography or x-ray. Patients were referred for rehabilitation based on physicians’ clinical judgment of whom may benefit from exercises to improve functional status. There was no separate protocol for the referrals, however, the patients must have presented post-COVID symptoms after 12 weeks from recovery.

The duration of therapeutic rehabilitation is a maximum of 21 days. The conditions for the provision of services, including those relating to medical personnel, equipment, and medical apparatus, meet the requirements set out in Annex 2 to the Regulation of the Minister of Health of 13 July 2020. The pilot project is carried out within the existing Pulmonary Rehabilitation Department of the SP ZOZ Specialist Hospital of the Ministry of the Interior and Administration in Glucholazy, Poland. The program is still ongoing.

The rehabilitation program is complex, yet based on currently recommended models of complex respiratory physiotherapy with adequate qualification rules including individualized treatment and rehabilitation program followed by an assessment of the effects in patients who recovered from COVID-19 with complications. Existing clinical observations allowed for modifications of the applied rehabilitation program based on the adequate qualification of the patient and appropriate selection of rehabilitation model [19,20,21,22].

A brief characterization of the patients’ group in the pilot program, including gender, required care due to COVID-19 infection, and qualification for individual rehabilitation models (A–E), can be found below (Table 1, Figure 1).

## 3. Model of Qualification and Rehabilitation Program

The proposed model of patients’ qualification rests on well-tested and validated tools for functional assessment which allow the physiotherapist to adequately program physical exercises being the core of the therapy. The proposed battery of tests includes: exercise tolerance assessment (6-min Walk Test), dyspnea intensity assessment (Modified Borg Dyspnea Scale), functional fitness assessment (Fullerton Test), assessment of arterial blood saturation by pulse oximetry, lung ventilation function assessment via body plethysmography, assessment of long-lasting COVID-19 symptoms (authors’ questionnaire) and patient’s mental health condition (Hospital Anxiety and Depression Scale).

### 3.1. Exercise Tolerance Assessment

Exercise tolerance is evaluated with the use of a 6-min Walk Test (6MWT), also known as a walk test or a corridor test. It allows global examination of the respiratory system, cardiovascular system, and neuromuscular system functions through the assessment of an aerobic capacity and endurance. Designed for geriatric and cardiopulmonary patients, but widely used in varied adult and pediatric populations. Can be conducted in any place, both indoors and outdoors, with a straight distance of 30 m on a hard, flat surface. Walking distance should be marked with cones for better orientation of the patient and include clear indications for every 3 m. The equipment required to conduct the test includes a stopwatch, pulse oximeter, blood pressure monitor, oxygen supply, and a questionnaire for subjective assessment of the level of dyspnea and tiredness (Modified Borg Dyspnea Scale). Before the test, the patient should be asked to rest in the sitting position for 10 min. Intensive physical effort is prohibited 2 h prior. The test should also be conducted on an empty stomach or after a light meal. Blood pressure, pulse, saturation, and dyspnea level should be checked directly before the beginning of the test. The patients are instructed to walk at their own speed and not to run; it is allowed to change their pace, accelerate, or slow down depending on their abilities, they can slow down or stop if they feel tired or short of breath. The goal of the 6MWT is to cover the longest possible distance rather than achieve the shortest time.

After the test, energy expenditure expressed in metabolic equivalent units (MET) is calculated. One MET equals average energy expenditure at rest (sitting position) and is equivalent to oxygen use at rest in sitting position by a person weighing 70 kg, i.e., 3.5 mL O_2_ × kg^−1^ × min^−1^. MET is determined based on the covered distance, calculated into walk speed, with the following equation: distance × 10/1000, e.g., 500 m = 5.0 km/h.

The previously existing formula for calculating MET based on the 6MWT was intended for people with severe limitations in exercise capacity, excluding patients who cover a distance of more than 300 m (the results obtained were biased). Modified formula based on the estimation developed on COPD patients (with various exercise impairments) can be used for all patients who require a quick, simple, and effective assessment of their exercise capacity, including those who can cover more than 300 m in the 6MWT [23]. The formula was developed in order to qualify homogeneous groups of patients and load them appropriately with physical effort. It has been tested and proven in COPD patients, but can and should be used in patients after COVID-19 with limited exercise capacity.

The figure below helps to read MET values: walk speed in km/h on the horizontal axis, and MET values on the vertical axis (Figure 2).

### 3.2. Dyspnea Intensity Assessment

Dyspnea intensity is assessed with the use of the subjective 10-point Modified Borg Dyspnea Scale (MBS). MBS is a safe and simple method, used in hospital conditions. ‘0’ at the top of the scale represents no perceived exertion while ‘10’ at the bottom represents the most intensive exertion ever experienced or possible to imagine. To determine the intensity, the patient selects the value which best describes the exertion they experience at the moment (Table 2).

### 3.3. Functional Fitness Assessment

Functional fitness is assessed with the use of one of the components of the Fullerton test known also as the Senior Fitness Test (SFT). It is a safe and simple test comprised of 6 components measuring the strength of the upper and the lower body, aerobic endurance, motor coordination, and balance. For qualification purposes, the ‘30-s Chair Stand’ test was used.

The patient sits on the chair, back to a wall, with feet flat on the floor and arms folded across their chest. On the ‘Start’ command, the patient stands up to assume an upright position and sits back on the chair, stands up, and sits back down again. The number of cycle repetitions within 30 s represents the test result. The ‘30-s Chair Stand’ test assesses muscle endurance in the lower body part used for getting up, walking, climbing stairs, and maintaining body balance.

Based on exercise tolerance and energy expenditure expressed in MET in the 6MWT, dyspnea assessment (10-pointMBS) and functional fitness assessment with the use of Fullerton test (‘30-s Chair Stand’ test) patients are qualified for adequate rehabilitation models A, B, C, D, or E (Table 3 and Table 4). The results of the exercise tolerance test are also used to determine the appropriate exercise load for each patient individually.

### 3.4. Assessment of Arterial Blood Oxygen Saturation

Arterial blood oxygen saturation is another parameter assessed in the process of qualifying patients for the rehabilitation program. The level of hemoglobin saturation with oxygen in arterial blood is tested with the pulse oximetry method (spectrophotometry transmission) during the 6MWT. Saturation norm is approx. 95–99%.

### 3.5. Lung Ventilation Function Assessment

Another parameter assessed in the qualification process is the Total Lung Capacity (TLC) index measured in body plethysmography, also known as body plethysmography. It is a lung function test used for the assessment of the total amount of air in the lungs as a certain amount of air remains in the lungs that cannot be exhaled. This is called residual volume (RV) and cannot be measured during spirometry. Plethysmography also allows us to determine the resistance to which air meets going through the respiratory tract. Obturation degree can be measured indirectly in this way. The lowest limit for TLC is set at the 5 percentile in reference population and equals the predicted value. The percentile is calculated with the function of cumulated probability for normal distribution and is equivalent to a given deviation from the predicted value. The result expressed in percentiles shows what percentage of the healthy population (respective to sex, age, and height) scores lower than the patient tested (e.g., 50 percentile means that the result of the patient tested represents the exact average for the healthy population). The principles for interpreting the results presented are based on the Recommendations of the Polish Society of Lung Diseases. The limits of the norm are determined arbitrarily on the basis of the probability theory and allow a certain margin of error, therefore the borderline results are always interpreted with great caution.

In case of saturation below 92% and/or TLC < 5 percentile, the rehabilitation model is lowered by one group.

### 3.6. Assessment of Additional Symptoms

Given the deficits which occur long after the infection, complex therapy focused on restoring the function of skeletal muscles, as well as physical and mental aspects is recommended. Apart from symptoms from the respiratory system, COVID-19 convalescents frequently experience weakened muscle strength, coordination and balance disorders, memory and concentration disorders, anxiety, and depression symptoms. The qualification for rehabilitation procedure, therefore, includes the results of the authors’ questionnaire on the subjective presence of additional post-COVID symptoms, Hospital Anxiety and Depression Scale (HADS), and record of the assessment of PCFS syndrome (Post COVID-19 Functional Status) [24]. Treatment in separate rehabilitation models is adjusted to the range and intensity of reported symptoms. It may result in selecting a lower rehabilitation model.

### 3.7. Rehabilitation Program

The proposed post-COVID-19 rehabilitation program includes five different models of therapeutic activities (A, B, C, D, and E) based on the individual exercise capacity of the patient. The models differ mainly in the intensity of exercise, which is determined for each patient also based on additional COVID-related symptoms and comorbidities. The specific abilities and needs of the patient should be taken into account, which are analyzed both on the basis of the results obtained in screening tests, anamnesis, and previous medical history.

Models A, B, and C consist of physical efficiency training on a cycle ergometer, walking training, resistance training, general fitness and circuit training, breathing exercises, procedures used to directly remove bronchial secretions, inhalations, and relaxation techniques. While the elements of the rehabilitation program for each of the models are the same, the difference is related to the intensity of training, starting from 80% of the submaximal heart rate for model A to 60% of the submaximal heart rate for model C. Models D and E consist of breathing exercises, general fitness exercises, circuit training, procedures used to directly remove bronchial secretions, inhalations, and relaxation techniques. The elements included in the rehabilitation program for both models are the same, the difference results from the intensity of the training sessions: heart rate increase during exercises by 20–30% in relation to the heart rate at rest in model D while in model E exercises are conducted in a sitting position on a chair with a heart rate increase during exercises by 20–30% in relation to the heart rate at rest.

Particularly important in the program is the use of new technologies, including breathing exercises with biofeedback, in order to improve respiratory dysfunction; relaxation in virtual reality (VR) conditions; interactive exercises combining cognitive exercises and physical exercises in VR focused on improving concentration, perception, memory, understanding and efficiency of communication, balance and coordination, strengthening the muscles of the lower and upper limbs, improving movement control, improving reaction time. Thanks to VR systems, patients can use sports facilities, such as a volleyball court, a climbing wall, a tennis court, and an obstacle course in hospital conditions.

The rehabilitation program also uses a high-intensity electromagnetic field to stimulate diaphragm contractility, intercostal muscles, and respiratory auxiliary muscles; stochastic resonance to normalize muscle tone, improve static and dynamic balance and coordination, stimulate deep sensory perception, and restore gait dysfunction; microcirculation therapy and vascular therapy.

The individual elements for each model in the presented rehabilitation program have been described and published in a separate article [25].

The proposed activities are carried out both individually and in groups depending on the patient’s current condition, postoperative complications, observed symptoms, their scope, and severity with a special focus on disorders of the musculoskeletal and nervous systems, as well as disorders of mental. It should be remembered that during the rehabilitation process, the patient’s health should be assessed regularly not only in order to assess the progress, but also for possible deterioration and undesirable effects.

## 4. Discussion

Patients who recovered from COVID-19 and require rehabilitation due to varied health impairments should be provided an opportunity to participate in an individualized, complex rehabilitation program starting from acute and early post-acute care and being continued in the post-acute and long-term rehabilitation phase. It is recommended to offer out-patient and in-hospital rehabilitation procedures depending on the type and persistence of symptoms and dysfunctions [26,27]. Data presented in a Swedish study from 2021 suggest that a substantial minority of patients with severe neurocognitive, cardiopulmonary, and/or sensorimotor disorders qualify for and require complex multi-profile rehabilitation, which can be delivered only by specialized centers [28]. The scale and specificity of the condition make the type of treatment, referral criteria, forms, and methods of treatment challenging in terms of availability of services and used protocols of therapy. Published research shows that regardless of the time since discharge from the hospital or recovery from infection, the QOL of patients is significantly affected. Emerging attempts of identification of low-QOL’s risk factors indicate female sex, older age, the presence of co-morbidities, ICU admission, prolonged ICU stay, and mechanical ventilation [29]. Long-term negative consequences of COVID-19 on health-related quality of life (HR-QoL), impaired mobility and limited ability to carry out activities of daily living is especially well documented among geriatric patients [30].

Taking into account both the previous experience related to physiotherapy in respiratory system diseases as well as the occurring and described post-COVID-19 symptoms lasting more than 12 weeks and requiring rehabilitation in hospital conditions among survivors, a simple, point-coded qualification model based on the available, verified and described in the literature on the subject tests has been developed. The basic assumption was a balance between a holistic approach to the highly unspecific symptoms, yet underlying functional impairments of patients, the availability of therapeutic services in healthcare facilities, capacity of sharing different therapeutic approaches within the multidisciplinary team, and the selection of a set of diagnostic tools that would be understandable and known to physiotherapists. The novelty in the proposed qualification is the scoring system. It is based on the results of the described tests, enabling the appropriate assignment of the patient to the specific post-COVID rehabilitation models, differing in the intensity of physical exertion and specific treatments adapted to the symptoms from the individual ranges mentioned above [31]. The call to prioritize rehabilitation for the medium- and long-term consequences of COVID-19 was shared by World Health Organization requiring at the same time a more systemic approach and continuous work on data gathering [32]. Hospital-based rehabilitation is cost-effective but is limited by the limited availability of staff [33].

Physical exercises treated and dosed like medicine are the basis of physical therapy. Adjusting the effort to the individual abilities of the patient enables an appropriate and planned qualification process based on the original, modified determination of energy expenditure expressed in MET and an easy and quick assessment of indicators of other elements of the functional assessment, enabling self-assessment of physiotherapeutic methods. Published papers point out that early rehabilitation for COVID-19 hospitalized patients was associated with lower in-hospital mortality even with a rather low dosage of exercises [34]. Lack of physical activity is proven to be connected with more severe course of infection, while physical exercises are beneficial for post-COVID patients, also when delivered through new technologies [35]. Positive health-related effects of physical activity have been previously presented in the WHO guidelines on physical activity and sedentary behavior published in 2020 [36,37]. In addition, in healthy people, exercise increases hippocampal volume and blood flow, stimulates neurogenesis, modulates synaptic plasticity, and increases growth factors such as BDNF, which are involved in optimizing brain function, which may be applicable to mental health challenges and neurocognitive impairment observed among the post-COVID patients [38].

The presented process of qualification of adult post-COVID-19 patients to the rehabilitation program consisting of 5 different therapeutic models, has some limitations. First of all, conducting a full functional examination in accordance with the described qualification and then continuing a comprehensive rehabilitation program requires housing facilities, administrative support, and a large, multidisciplinary team of health care workers. The size of the center does not determine the possibility of introducing the presented qualification, but it can significantly affect the effectiveness and the number of patients who can participate in the rehabilitation program at the same time. Modifications for the smaller centers and outpatient care facilities, as well as further development of therapeutic activities using new technologies, should be considered. The second significant limitation of the proposed qualification is the lack of a comprehensive assessment of the mental state conducted by specialized clinicians. Mental health may affect both the course of the underlying disease and the effects of the rehabilitation process.

The methodology presented in the paper is associated with new perspectives regarding the health care for adult post-COVID-19 patients. As it is universal in terms of the assessment of varied needs related to in-/outpatient rehabilitation and subsequently, further therapeutic decisions, thus can be introduced outside hospital-based rehabilitation units. It allows a functional assessment of the patient’s post-COVID-19 fitness, so it can fulfill more than just the role of qualification for rehabilitation models.

Additionally, the presented qualification indicates the need for the usage of a modified formula for calculating MET based on the 6MWT in order to qualify homogeneous groups of patients after COVID-19 with limited exercise capacity and load them appropriately with physical effort. A cardiac stress test is time-consuming and costly in comparison, equipment-related requirements may make it impossible to perform in smaller therapeutic centers.

## 5. Conclusions

The proposed qualification model, confirmed by the previous theoretical knowledge and clinical experience, is, therefore, an attempt to present a transparent, uniform functional assessment of the patient before starting the rehabilitation process. Such a qualification enables appropriate planning and adaptation of therapeutic activities to the needs of patients, as well as the subsequent broad assessment of therapeutic effects after the end of the program.

The wide application of the unified qualification model also enables a more universal functional assessment of post-COVID-19 patients, which will allow further better and more directed healthcare. Structured clinical observation over a longer period of time may also give a better understanding of the long COVID-19 symptoms.

However, for holistic patient care, consideration should be given to introducing regular mental status assessments.

## Figures and Tables

**Figure 1 ijerph-19-10450-f001:**
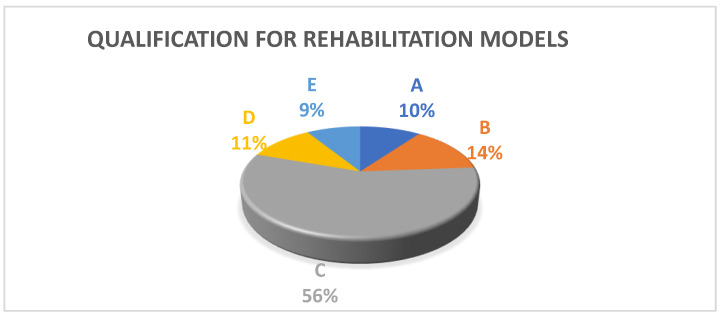
Qualification for five models of rehabilitation (A–E) in the period of 1 September 2020–17 June 2022.

**Figure 2 ijerph-19-10450-f002:**
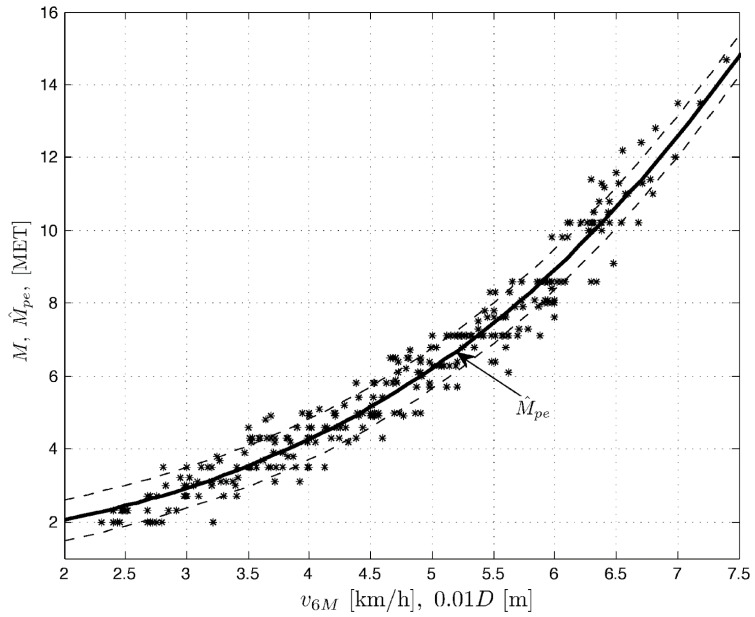
MET values based on the covered distance in the 6MWT and calculated into walk speed [km/h] [23].

**Table 1 ijerph-19-10450-t001:** Brief characterization of the patients participating in the post-COVID-19 rehabilitation program in the period of 1 September 2020–17 June 2022.

Patients			Required Care		
Sex	Number	Age	ICU	Hospital Treatment	Treatment at Home
Female	1335	M 63 MIN 27 MAX 92	198	498	639
Male	1424	M 62 MIN 26 MAX 96	234	510	680
Sum	2759	M 63 MIN 26 MAX 96	432	1008	1319

**Table 2 ijerph-19-10450-t002:** Modified Borg Dyspnea Scale.

How Difficult Is Your Breathing Now?	
0	Nothing at all (rest)
0.5	Very, very slight (just noticeable)
1	Very slight
2	Slight
3	Moderate
4	Somewhat severe
5	Severe
6	
7	Very severe
8	
9	Very, very severe (almost maximal)
10	Maximal

**Table 3 ijerph-19-10450-t003:** Qualification Card for the post-COVID-19 Rehabilitation Program used by physical therapists, part 1.

Test	Qualification					Points
Exercise test (MET)	>7	>5–7	>3–5	≤3	No test	…
Dyspnea (10-pointMBS)	0.5–1	2–3	4	5–6	>7	…
Functional fitness (SFT ‘30-s Chair Stand’ number of repetitions)	>15	12–14	9–11	6–8	<6	…
Points	5 pts	4 pts	3 pts	2 pts	1 pt	Points in total:…

**Table 4 ijerph-19-10450-t004:** Qualification Card for the post-COVID-19 Rehabilitation Program used by physical therapists, part 2.

Rehabilitation Models					
Rehabilitation Model	Total Pts	Total Pts	Total Pts	Total Pts	Total Pts
	15–13	12–10	9–7	6–4	3
	A	B	C	D	E

## Data Availability

Not applicable.

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
