# Peer review of "Proprietary Model of Qualification for In-Hospital Rehabilitation after COVID-19"

_ijerph, 2022, doi:10.3390/ijerph191610450_

Round 1
Reviewer 1 Report
In this paper Szczegielniak et al. presents the model used for patients’ qualification into five different levels of rehabilitation. The model is based on few easily available and reproducible tools (6MWT, Borg scale, ’30-Second Chair Stand’ test, blood oxygen saturation, TLC). The possible activities used during the rehabilitation program are briefly summarized. This model has the aim to standardize possible studies on the efficacy of rehabilitation programs in long COVID-19 patients.
The overall presentation of the paper is clear, it surely responds to its aim of explaining how to qualify patients with long COVID-19 with accessible tools. I think that the paper could be published only with an English revision to be easily readable and with some minor revision. However, I want to rise some other issues, hoping to add some more scientific interest in the paper.
Major revisions:
- The MET non-linear estimation using 6MWT has been validated by the authors in another paper using a COPD cohort; firstly, it should be stated in the paper to clarify how the authors quantified the MET in long-COVID-19 patients. Second, are the authors sure that the MET quantification derived from 6MWT could be extended from a specific population such as COPD patients to a general population? Even if not essential for this paper, it should be considered for further analysis;
- Paragraph 3.7 explaining the rehabilitation program should be revised, it is hard to understand the different approaches possible for the rehabilitation. I think this is the most interesting part of the paper, so different approaches should be better divided in paragraphs and explained. If possible, a revised form or a copy of the table in the cited paper [26] should be added to better understand the rehabilitation program;
- Even if not central in this paper, a brief presentation of the evaluated patients could be of interest. I’d appreciate if the authors will add a short presentation with only few information such as sex, age, ICU admission (etc.) and categorization into the 5 groups of rehabilitation, to understand how the long covid-19 affects different patients.
Minor revisions:
- Introduction section is well written, but it could be shortened to be easier to be read and more focused on the aim of the paper;
- Tables and figures should be readable as stand alone, I belive that the titles of all the figures and tables could be more explicative of the content (for example line 139-140 could be added as the description of fig. 1); furthermore, tables should be numbered in order of appearance (there are 2 different table 1);
- Authors should pay attention on abbreviations, even if some of them are standardized, they should always be expressed (e.g. ICU for intensive care unit), also if something has been abbreviated once, it should be used always abbreviated (e.g VR for virtual reality in results section).
Author Response
We would like to thank you immensely for the time and effort to revise our paper. We hope that the changes we have introduced to the manuscript have significantly increased the quality of this work. We have conducted major revision of the language used in the manuscript and considered all of the suggestions.

Reviewer 2 Report
1) The proposed qualification model for the rehabilitation of patients in the post-COVID-19 stage is significant and benificial to the intersted readers.
2) It should be with more interest if behavior of MET is reached deterministically or stochastically in view of predicting future.
3) It will be interesting to use \Labals and \eqref in order to access easily to references.
4) page 7, Why choosing normal distribution instead of other ones?
5) It is preferable to adress Testing hypothesis method to confirm obtainig results in order to get a consistent scientific soundness in the paper.
6) The conclusion should cover all achievements in the paper.
What is the limitations of the proposed qualification program.
There is any Perspectives to announce in this context.
Author Response
We would like to thank you immensely for the time and effort to revise our paper. We hope that the changes we have introduced to the manuscript have significantly increased the quality of this work. We have conducted major revision of the language used in the manuscript and considered all of the suggestions.
Please see the attachment.

Reviewer 3 Report
In this study, the authors ( PROPRIETARY MODEL OF QUALIFICATION FOR IN-HOSPITAL REHABILITATION AFTER 2 COVID-19 ). that patients who recovered from COVID-19 and require rehabilitation due to varied impairments should be provided an opportunity to participate in an individualized, complex rehabilitation program starting from acute care and being continued in the post-acute and long-term rehabilitation phase. The proposed qualification model for the post-COVID rehabilitation program allows to introduce adequate qualification rules followed by much needed assessment of the health effects.
For the international readership, the topic is really interesting, as adequate qualification rules, practical and theoretical implications are mentioned.
In the introduction and abstract I suggest to add some information of the situation of COVID-19 and how did the Polish government deal with it and the cost of treatment.
I was hoping to see a further prospective clinical study of the effects of the used could better support the claims.
The discussion is short, the figures could be better explained in the main text. I suggest add some measures of validity and reliability. Moreover, add the strengths, limitation, and a table for basic characteristics to have a general view.
Author Response

(The authors gave the same response as above.)

Round 2
Reviewer 1 Report
I really appreciate the effort of the author to respond to my review, and I believe the paper is suitable for pubblication in this present form.